# Altered Mucus Barrier Integrity and Increased Susceptibility to Colitis in Mice upon Loss of Telocyte Bone Morphogenetic Protein Signalling

**DOI:** 10.3390/cells10112954

**Published:** 2021-10-29

**Authors:** Vilcy Reyes Nicolás, Joannie M. Allaire, Alain B. Alfonso, Dianne Pupo Gómez, Véronique Pomerleau, Véronique Giroux, François Boudreau, Nathalie Perreault

**Affiliations:** Département d’Immunologie et Biologie Cellulaire, Faculté de Médecine et des Sciences de la Santé, Université de Sherbrooke, Sherbrooke, QC J1E 4K8, Canada; Vilcy.Reyes.Nicolas@USherbrooke.ca (V.R.N.); jallaire@bcchr.ca (J.M.A.); Alain.Barbaro.Alfonso.Chaviano@USherbrooke.ca (A.B.A.); Dianne.Pupo.Gomez@USherbrooke.ca (D.P.G.); Veronique.Pomerleau@USherbrooke.ca (V.P.); Veronique.Giroux@Usherbrooke.ca (V.G.); Francois.Boudreau@USherbrooke.ca (F.B.)

**Keywords:** inflammatory bowel diseases, cellular microenvironment, wound healing, goblet cells, *O*-glycosylation

## Abstract

FoxL1^+^-Telocytes (TC^FoxL1+^) are subepithelial cells that form a network underneath the epithelium. We have shown that without inflammatory stress, mice with loss of function in the BMP signalling pathway in TC^FoxL1+^ (*BmpR1a*^ΔFoxL1+^) initiated colonic neoplasia. Although TC^FoxL1+^ are modulated in IBD patients, their specific role in this pathogenesis remains unclear. Thus, we investigated how the loss of BMP signalling in TC^FoxL1+^ influences the severity of inflammation and fosters epithelial recovery after inflammatory stress. *BmpR1a* was genetically ablated in mouse colonic TC^FoxL1+^. Experimental colitis was performed using a DSS challenge followed by recovery steps to assess wound healing. Physical barrier properties, including mucus composition and glycosylation, were assessed by alcian blue staining, immunofluorescences and RT-qPCR. We found that *BmpR1a*^ΔFoxL1+^ mice had impaired mucus quality, and upon exposure to inflammatory challenges, they had increased susceptibility to experimental colitis and delayed healing. In addition, defective BMP signalling in TC^FoxL1+^ altered the functionality of goblet cells, thereby affecting mucosal structure and promoting bacterial invasion. Following inflammatory stress, TC^FoxL1+^ with impaired BMP signalling lose their homing signal for optimal distribution along the epithelium, which is critical in tissue regeneration after injury. Overall, our findings revealed key roles of BMP signalling in TC^FoxL1+^ in IBD pathogenesis.

## 1. Introduction

Intestinal inflammation has long been considered as a process in which effector immune cells destroy the mucosa and subsequently result in chronic inflammation when left unresolved [1]. Therefore, the mucosa is a target instead of a possible trigger for inflammatory bowel disease (IBD). Studies investigating the deregulation of non-hematopoietic mucosa cells in various phases of IBD have broadened the understanding of their involvement in IBD pathogenesis [2,3,4]. Thus, epithelial and mesenchymal cells also participate in IBD, from pathogen or damage recognition to recruitment of immune cells to the injury site, pathogen or damage elimination and, ultimately, resolution of inflammation [3,5]. Particularly, mesenchymal cells are strategically positioned between the epithelial and immune cell compartments and can consequently regulate epithelial functions and influence mucosal immune cells [5,6].

Gastrointestinal mesenchymal cells consist of a heterogeneous community of fibroblasts, myofibroblasts, FoxL1^+^-Telocytes (TC^FoxL1+^) and trophocytes, among others [3,6,7]. Myofibroblasts, trophocytes and TC^FoxL1+^ are especially bioactive with the production of growth factors, cytokines, chemokines and extracellular matrix (ECM) proteins, thus forming a microenvironment that supports epithelial and immune cell homeostasis [6,8,9,10]. Evidence suggests that myofibroblasts lack the heterogeneity to create the necessary gradients needed to support the ever changing epithelial functions along a vertical axis such as the crypt-villus in the small intestine or the crypt in the colon [7]. Conversely, TC^FoxL1+^ represent one of the mesenchymal sources of the gradients in addition to being of high relevance with regards to epithelial–mesenchymal interactions nesting directly under epithelial cells along the vertical axis [3,5,6,8]. The latter is of importance as TC^FoxL1+^ can differentially influence the epithelium depending upon their position along this axis [6]. As shown from the pericryptal mesenchymal cell population, TC^FoxL1+^ and trophocytes play a crucial role in the establishment of the stem cell niche as they are the major source of bone morphogenetic protein (BMP) inhibitors, WNT5a, WNT2B and R-spondin 3 [4,6,7,11]. In addition to their involvement in the stem cells niche, RNA-sequencing analysis in TC^FoxL1+^ also demonstrated expression of other key signalling molecules such as the BMP ligands, the BMPR1A receptor along with members of the Hedgehog and FGF signalling family. This indicates the full potential of TC^FoxL1+^ to also be involved in other cell processes such as epithelial determination, differentiation and functionality [6,7,12].

Recently, TC^FoxL1+^ have been reported to be modulated in patients with ulcerative colitis, suggesting that they are important in IBD pathogenesis. Moreover, signals from TC^FoxL1+^ help maintain the epithelial barrier integrity and immune homeostasis [3]. Physical and chemical barriers not only protect the colonic mucosa from inflammatory triggers contained in its lumen, but also prevent the harmful adhesion and invasion of microorganisms [13]. From the physical barrier, protection by the mucus layers has been of interest in IBD pathogenesis. Alterations in mucus-producing goblet cells or the composition, structure and thickness of the mucus itself directly affect epithelial protection [14]. The regulation of goblet cells is of particular interest in TC^FoxL1+^ as the latter has an essential role in the maintenance of the stem cell niche [4,6,11]. Because all colonic epithelial cells are derived from the same stem cell, the effect of TC^FoxL1+^ on the niche invariably affects epithelial cell-fate decisions and the maturation of progenitors into colonocytes or goblet cells [3,6,7]. However, the specific roles played by TC^FoxL1+^ in barrier integrity, goblet cell functionality, IBD initiation and resolution remain unclear.

Our previous work on genetically modified gut TC^FoxL1+^ (*BmpR1a*^ΔFoxL1+^ mice) has shown that disruption of the BMP signalling, an important pathway in gastrointestinal diseases [2,8,10,15], influences the microenvironment via the reprogramming of stroma into reactive mesenchyme, which subsequently initiates spontaneous neoplasia in the epithelium [8]. In addition, *BmpR1a*^ΔFoxL1+^ mice revealed that mutant mice developed colonic dysplastic regions with substantial early onset stromal changes, well before polyposis. With time, the enlarged mesenchymal compartment of *BmpR1a*^ΔFoxL1+^ mice presented immune cells infiltration but this never led to a spontaneous inflammatory flare [8]. This study revealed the potential of signalling-impaired TC^FoxL1+^ in initiating gastrointestinal diseases such as neoplasia. Although it is clear that mesenchymal cells can contribute to IBD pathogenesis [3,5,8,9], the precise effect of TC^FoxL1+^ in IBD remains unexplored.

Therefore, in this study, we explored the role of BMP-TC^FoxL1+^-associated signalling in IBD susceptibility and wound healing, as well as its effect on colonic physical barrier homeostasis in mice with conditionally inactivated *BmpR1a* in TC^FoxL1+^. Consistent with the important regulatory role of TC^FoxL1+^ in the maintenance of epithelial homeostasis, our findings indicate that defective TC^FoxL1+^ are key contributors in IBD susceptibility and limit the tissue wound healing ability along with resolution of inflammation. We demonstrate that for the maintenance of tissue functionality following injury, TC^FoxL1+^ population localisation along the colonic epithelial axis is critical to resolve and repair the inflammatory insult. Overall, we show that BMP signalling plays a critical role in the optimal functionality and homeostasis of TC^FoxL1+^, thereby affecting epithelial injury and repair.

## 2. Materials and Methods

### 2.1. Animals

The transgenic line C57BL/6J *FoxL1*-Cre was provided by Dr. Kaestner [16] and the C57BL/6J -*BmpR1a^fx/fx^* mouse was gifted by Dr. Mishina [17]. *BmpR1a*^ΔFoxL1+^ conditional knockout mice were generated as previously described [8,10].

### 2.2. Induction and Assessment of DSS-Induced Colitis and Recovery

Colitis was induced in post-natal 90-day-old *BmpR1a*^ΔFoxL1+^ and control mice using dextran sulphate sodium (DSS; MW: 35,000–50,000) as previously described [2]. For the acute colitis experiment, mice were fed with 3% (*w*/*v*) DSS water (*n* = 10 per sex and group) ad libitum for 7 days. For the recovery group, mice were fed with 3% (*w*/*v*) DSS water (*n* = 12 per sex and group) ad libitum for 5 days, followed by tap water for 14 days. Mice were assessed for disease activity using the modified criteria by Cooper et al. [18] and histological scoring was based on Dieleman et al. [19].

### 2.3. Histological Analysis and Grading of Colitis

Colons from 90-day-old mice were fixed as previously described [8]. Histological scoring was blindly performed on four colons from different mice per group using 12 high-field images of H&E-stained sections. To analyse the mucus barrier, tissues were fixed in Carnoy’s solution as previously described [20]. Tissues were sectioned and stained with H&E or alcian blue following standard protocols [2,8].

### 2.4. Intestinal Permeability Assay

Permeability assay was performed in 90-day-old *BmpR1a*^ΔFoxL1^ and littermate control mice. Both groups were fasted for 16 h and 150 µL of 80 mg/mL of 4-kDa fluorescein isothiocyanate (FITC)-dextran (Sigma-Aldrich, St. Louis, MI, USA) was applied orally at a single dose. Mice were anaesthetised after 4 h and serum was collected. Fluorescence was measured spectrophotometrically (Infinite M200 PRO, Tecan, Crailsheim, Germany) in 96-well plates (excitation: 492 nm, emission: 525 nm). FITC-dextran concentrations were calculated using a standard curve prepared in serum ranging from 0 to 80 µg/mL 4-kDa FITC dextran.

### 2.5. Electron Microscopy

Colons from post-natal 90-day-old *BmpR1a*^ΔFoxL1+^ and control mice were fixed and sectioned as previously described [2]. Transmission electron microscopy images were digitally coloured in blue using Adobe Photoshop CC 2017.

### 2.6. Immunofluorescence and Fluorescence In Situ Hybridisation

Immunofluorescence staining for TC^FoxL1+^ population was performed as follows; slides were immersed in 0.01 M citric acid buffer (pH 6.0) and microwaved to boil for 6 min for antigen retrieval, then cooled, washed in PBS and incubated for 40 min at room temperature in blocking solution (1% Gelatin from cold water fish skin, 2% BSA, 0.2% Triton X-100 in PBS). Two first primary antibodies anti-Gli1 (1:500, NB600-600, Novus Biological, CO, USA) and anti-PDGFRα (1:150, AF1062, R&D System, MN, USA) were simultaneously diluted in the blocking solution and incubated overnight at 4 °C in a moist chamber. After washing with PBS, Alexa 594 (anti-goat) and Alexa 647 (anti-rabbit) secondary antibodies were diluted in blocking solution, applied to the slides and incubated for 1 h at RT. A second blocking step was performed incubating the slides for 40 min at room temperature in the blocking solution described above. Then, slides were incubated 1 h at RT with anti-CD34 (1:100, ab81289, Abcam, Cambridge, UK) diluted in the blocking solution, followed by 1 h with Alexa 488 (anti-rabbit) at RT. Nuclei were stained with DAPI. Images were captured on a confocal Microscope Zeiss LSM 880 2 photons. Telocytes population was analysed using Fiji ImageJ v 2.1.0, from high-powered fields in a blinded manner on an average of 10 independent fields of the whole colon per animal (*N* = 6 per group). Staining on frozen sections was performed as followed: tissues were fixed in 100% Ethanol for 15 min at −20 °C [2,8,10,15]. For all other antibodies, immunofluorescence was assessed as previously described [2,8,10]. The following antibodies were used at the indicated dilutions: Anti-MUC2 (1:200, sc-15334, H-300, Santa Cruz); Anti-JAM-A (1:25, 36–1700, Zymed laboratories, San Francisco, CA, USA) anti-Claudin 1 (1:50, 51–9000, Zymed laboratories, San Francisco, CA, USA), anti-Claudin 2 (1:80, 51–6100, Invitrogen, Waltham, MA, USA), anti- ZO-1 (1:50, 61–7300, Zymed laboratories, San Francisco, CA, USA); co-staining for myofibroblast population, anti-vimentin (1:100, 5741S, Cell signaling, MA, USA) and anti-αSMA (1:5000, A2547, Sigma-Aldrich, St. Louis, MI, USA)); Alexa 488-conjugated anti-rabbit (1:300, 4412S, Cell Signalling, MA, USA); Alexa 594-conjugated anti-rabbit (1:300, 8889S, Cell Signalling, MA, USA); FITC-labelled Anti-rabbit (1:300, FI-1000, Vector); Alexa 647-conjugated anti-rabbit (1:300, A21443, Invitrogen, Waltham, MA, USA); Alexa 594-conjugated anti-goat (1:300, A11058, Invitrogen, Waltham, MA, USA) and Alexa 594-conjugated anti-mouse (1:300, 8890S, Cell Signalling, MA, USA). Images were captured on a Microscope Zeiss Axioscope 5. Carnoy-fixed colon sections were hybridised with a Cy3-coupled bacterial 16S rRNA probe (EUB338). A Cy3-coupled nonsense probe (NS_EUB338) was used as the control for non-specific binding. Probe sequences for fluorescence in situ hybridisation were CY3_EUB338_SENSE 5′-/5Cy3/GCTGCCTCCCGTAGGAGT-3′ and CY3_NS_EUB338 5′-/5Cy3/CGACGGAGGGCATCCTCA-3′. Nuclei were stained with DAPI. Images were captured on a Leica DM2500 Optigrid.

### 2.7. Analysis of Post-Translational Modifications in Goblet Cell and Mucins

Slides were incubated with lectin probes: FITC-labelled Ulex europaeus agglutinin I (UEA-1; 1:750, L9006, Sigma-Aldrich, St. Louis, MI, USA) and Cy3-labelled Sambucus nigra lectin (SNA; 1:750, CL-1303, Vector Laboratories, ON, CA). For biotinylated Maackia amurensis lectin II (MAL II; 1:750, B-1265, Vector Laboratories, ON, CA), the sections were treated with avidin and biotin. The naive streptavidin protein Texas red (1:200, ab136227, Abcam, Cambridge, UK) was used to label the biotinylated MAL-II. FITC-labelled peanut agglutinin (PNA; 1:750, FL-1071, Vector Laboratories, ON, CA) was evaluated as described previously, [21] with and without enzymatic digestion and chemical pretreatment.

### 2.8. Quantification of Cell Number, Cell Distribution, Vesicle and Goblet Cell Post-Translational Modifications

The presence of goblet cells was analysed using alcian blue-stained sections from low-powered fields of well-oriented colonic cross-sections in a blinded manner on an average of 10 independent fields of the proximal, middle and distal colon per animal (*N* = 4 per group). Goblet cell vesicles were analysed from 10 pictures from transmission electron microscopy images. Using Fiji ImageJ v 2.1.0 they were divided into clear, light grey and dark grey and then quantified (*N* = 3 per group). For lectins and MUC2 immunofluorescence, we measured the corrected total cell fluorescence (CTCF) by applying the following formula CTCF = Integrated Density—(Area of selected cell X Mean fluorescence of background readings), using Fiji ImageJ v 2.1.0. TC^FoxL1+^ were labelled according to the most recent consensus in the literature as described in Section 2.6 [7]. Triple immunostaining for PDGFRα, Gli-1 and CD34 was performed, and TC^FoxL1+^ were identified as PDGFRα^+^/Gli-1^+^/CD34^−^. TC^FoxL1+^ number was averaged from ten different image fields per mouse in each condition (naive, acute, recovery). TC^FoxL1+^ distribution was also evaluated from these images, where the colon mucosa was separated in 3 different zones, based on epithelial cell identity: stem cells and progenitor cells, transit-amplifying and differentiated zones. TC^FoxL1+^ were assigned to either one of the zones in all 3 conditions (naive, acute, recovery). Multiple images for each mouse were evaluated and the relative frequency of distribution in percentage was then calculated (zone SC-P + zone TA + zone D = 100%). For myofibroblasts, total number of double-positive cells for Vimentin/αSMA was counted using Fiji ImageJ v 2.1.0, from high-powered fields in a blinded manner on an average of 10 independent fields of the distal colon per animal (*N* = 4–6 per group).

### 2.9. RNA Extraction and Gene Expression

Total RNA was isolated from the colon of control mice and the colonic dysplastic regions of *BmpR1a*^ΔFoxL1+^ mice using a Totally RNA extraction kit (Thermo Fisher Scientific, MA, USA) [8,12]. Epithelial total RNA was isolated as previously described [22]. Reverse transcription and quantitative RT-qPCR were performed as previously described [2,8,10]. For every qPCR run, a no-template control was performed for each primer pair, each of which was consistently negative. For every qPCR run, specific PCR primer sequences were used to detect the presence of Hes1; F: 5′-CTTCAATTGGTCCTGTCCAT-3′, R: 5′-CCTGTCTGGGAGGATCAAAA -3′; Math1, F: 5′-CGATGATGGCACAGAAGGA-3′, R: 5′-GGGGAAAACTCTCCGTCACT-3′; Gfi1, F: 5′-ATG TGC GGC AAG ACC TTC-3′, R: 5′-TCCGAGTGAATGAGCAGATG-3′; Klf4, F: 5′-CAGTATACATTCCGCCACAGC-3′, R: 5′-TCCGAGTGAATGAGCAGATG-3′; Spdef, F: 5′-GCCTGGATGAAGGAGAGGAC-3′, R: 5′-GGCTTGAGCAGCAGTTCTTT-3′; Muc1; F: 5′-GGACACCTACCATCCTATGAG-3′, R: 5′-CTTGCTCCTACAAGTTGGCAG-3′; Muc2, F: 5′-CTTCAATTGGTCCTGTCCAT-3′, R:’5′-TGGCTAAACACGCTTCTCCT-3′; Muc3, F: 5′-CTTCCAGCCTTCCCTAAACC-3′, R: 5′-TGGCTAAACACGCTTCTCCT-3′; Muc4, F: 5′-ATCCACTATCTGAACAACCAGC-3′, R: 5′-GTAGCCATCACATGTGAAGTAC-3′; Fut2, 5′-GAGTCAAGGGGAGGGAGAAC-3′ R: 5′-CCAGGGCTACAGAAGTGGAC-3′; Tff3, F: 5′-GCTGCCATGGAGACCAGA-3′, R: 5′-GAGCCTGGACAGCTTCAAAA-3′; Fcgbp, 5′-TGGTTCTCAGGGGAAGACAC-3′, R: 5′-ACACAGGGCATCTTCCAATC-3′; Agr2, F: 5′-ACCGGCTCTACGCTTATGAA-3′ R: 5′-TCTGCAAGTCCACAGTGCTT-3′; Retlnb, F: 5′-CGCAATGCTCCTTTGAGTCT-3′, R: 5′-CCACAAGCACATCCAGTGAC-3′; Tbp, F: 5′-GGGGAGCTGTGATGTGAAGT-3′, R: 5′-GGAGAACAATTCTGGGTTTGA-3′.

### 2.10. Statistical Analysis

Statistical significance was calculated using the Mann–Whitney test in GraphPad Prism v8. * *p* < 0.05; ** *p* < 0.01; *** *p* < 0.001; **** *p* < 0.0001 were considered significant. Data are presented as the mean ± SEM. For the goblet cell vesicles counting and TC^FoxL1+^ distribution along the crypt axis, 2-way ANOVA was used as a statistical test. Data are presented as the mean ± SD.

## 3. Results

### 3.1. BmpR1a^ΔFoxL1+^ Mice Have Increased Susceptibility to Experimental Colitis and Delayed Wound Healing

Upon the induction of acute colitis with DSS (herein, acute phase), a reduction in body weight was observed in both male and female *BmpR1a*^ΔFoxL1+^ mice compared with the controls at day 5 (Figure 1A). After recovery from acute colitis (herein, recovery phase), there was no significant difference in weight loss in *BmpR1a*^ΔFoxL1+^ mice between the acute and recovery phases compared with the control (Figure 1B). Noticeably, a 60% death rate was observed in *BmpR1a*^ΔFoxL1+^ mice in the first two days of recovery. The body weights of the surviving mice were similar between both groups by day 11 until day 19 (end of the experimental period). We observed a significant 1.6-fold modulation in disease activity index (DAI) between the control and *BmpR1a*^ΔFoxL1+^ mice in the acute phase (Figure 1C). After completing the recovery phase, there was no significant difference in the DAI of surviving animals in both groups. Histological analysis results revealed normal colonic mucosa in naive control and *BmpR1a*^ΔFoxL1+^ mice (Figure 1D). Following the acute phase, crypt erosions, minor immune infiltration and a preserved epithelial layer were observed in the control, whereas complete loss of crypt architecture, strong influx of immune cells and total loss of the epithelial lining were observed in *BmpR1a*^ΔFoxL1+^ mice (Figure 1D). In the recovery phase, the controls showed nearly restored colonic mucosa compared with the surviving *BmpR1a*^ΔFoxL1+^ mice that still presented significant immune cell infiltration with some areas having crypt abscess or denuded epithelium (Figure 1D). H&E staining demonstrated significant differences between the control and *BmpR1a*^ΔFoxL1+^ mice in both acute and recovery phases. In both treatments, surviving *BmpR1a*^ΔFoxL1+^ mice had a 1.4-fold increase in their histological score after the acute (7 days) and recovery treatments (19 days) compared with the control mice (Figure 1E).

### 3.2. Defective Telocytes Lead to the Development of Compromised Mucus Layers and Abnormal Bacterial Infiltration

The loss of the epithelial barrier function has been reported as a cause of IBD [2,23]. However, impaired epithelial permeability was not observed in both groups (Figure 2A). The mucus layers serve as another protective mechanism in the epithelium against commensal bacteria as they form part of the physical barrier. Goblet cells are responsible for the production of mucus, thereby protecting the epithelial gut lining by forming the mucus barrier [24]. Under naive conditions, we observed no differences in goblet cell number along the colonic crypts in both groups, although we noticed a thinner and discontinuous barrier layer in *BmpR1a*^ΔFoxL1+^ mice (Figure 2B,C). Under acute phase, reduced goblet cells in the colonic crypt were observed in both groups; notably, in *BmpR1a*^ΔFoxL1+^ mice, an almost complete depletion of these cells was observed in the distal colon (Figure 2B,C). The partial erosion of the barrier layer occurred in all mice but was greater in *BmpR1a*^ΔFoxL1+^ mice (Figure 2C). In the recovery phase, we observed a partial restoration in goblet cell count and mucus layers in the control group, whereas delayed mucus layer restoration and low goblet cell density were observed in *BmpR1a*^ΔFoxL1+^ mice (Figure 2B,C).

The primary role of healthy mucus layers is to physically protect the epithelial surface against bacteria in the lumen. Given that *BmpR1a*^ΔFoxL1+^ mice showed discontinuous mucus layers, we hypothesized that commensal bacteria could directly interact with the epithelium. We found that under naive conditions, bacteria were only detected in the outer mucus layer in the control but within the barrier layer in close contact with the non-inflamed epithelium in *BmpR1a*^ΔFoxL1+^ mice (Figure 2D). Following the acute phase, the control mice presented with typical erosion of the barrier layer and increased bacteria near the epithelia. In contrast, a significant number of bacteria invading the inflamed colonic mucosa in *BmpR1a*^ΔFoxL1+^ mice were observed (Figure 2D). After recovery, the control mice had restored mucus layers and no bacteria were found in the sterile inner layer. Conversely, bacteria were still detected near the epithelial layer in *BmpR1a*^ΔFoxL1+^ mice (Figure 2D).

### 3.3. BMP-Associated Signalling in Telocytes Supports the Maturation of Colonic Goblet Cells

Lineage commitment of gut epithelial cells, along with goblet cell fate specification, involve the expression of transcription factors from the Notch signalling pathway and *Klf4* [25,26]. No modulation was found in the commitment genes *Hes-1* and *Atoh1*, or in specification genes *Spdef* and *Gfi1*, although a significant reduction was observed in the maturation gene *Klf4* (1.4-fold) (Figure 3A). Ultrastructural examination revealed normal morphology in secretory goblet cells, i.e., a basal nucleus localisation and apical mucin vesicle accumulation, in both groups (Figure 3B). Noticeably, the controls presented more heterogeneity in vesicles compare to *BmpR1a*^ΔFoxL1+^. Vesicles in *BmpR1a*^ΔFoxL1+^ mice were frequently fused together and less organised. We therefore quantified the number of clear, light grey and dark vesicles found in goblet cells in both *BmpR1a*^ΔFoxL1+^ and control mice (Figure 3C). We found no statistical differences for the clear vesicles content and a tendency to fewer dark vesicles in the goblet cells of *BmpR1a*^ΔFoxL1+^ mice. However, we observed a significant increase in light grey vesicles (1.7-fold) in mutant goblet cells when compared to controls (Figure 3C). The decrease in vesicle heterogeneity in *BmpR1a*^ΔFoxL1+^ mice led us to investigate mucin diversity and their structural components using RT-qPCR analysis. The mRNA levels of *Muc2* (3.01-fold), *Fut2* (1.4-fold), *Muc4* (2.4-fold) and *Argr2* (1.7-fold) were significantly increased in *BmpR1a*^ΔFoxL1+^ mice when compared to controls (Figure 3D). Electron microscopy analysis showed that in control mice, apical microvilli were coated with thick glycocalyx while *BmpR1a*^ΔFoxL1+^ mice showed very little coating (Figure 3E). Typical ultrastructural apical junctional complexes were observed in both groups (Figure 3E). Immunofluorescence against proteins involved in the tight junction complex, such as ZO-1, claudin-1, claudin-2 and JAM-A revealed no apparent change between both groups, suggesting the presence of a normal junctional complex in *BmpR1a*^ΔFoxL1+^ mice (Figure 3F).

### 3.4. BMP-Associated Signalling in Telocytes Affects Post-Translational Modifications of Mucins

We first investigated the distribution of MUC2 using immunofluorescence and observed its peculiar accumulation at the periphery of goblet cells in *BmpR1a*^ΔFoxL1+^ mice compared with its normal distribution in the control (Figure 4A–C).

UEA-I lectin recognises the Fucα1,2Galβ1,4 motif, and its lectin reactivity is distributed uniformly along the different regions of the colon. We observed that control mice possessed well-filled vesicles (Figure 4D); whereas *BmpR1a*^ΔFoxL1+^ mice had mucin vesicles with reduced fucosylated residues (Figure 4E,F). Sialic acid modifications in mucins were also investigated. SNA-lectin in the distal colon was only detected in the upper part of the crypts and exhibited a pattern similar to the one observed in MUC2, i.e., accumulation of goblet cells in the periphery in *BmpR1a*^ΔFoxL1+^ mice compared with the control (Figure 4G–I). MAL-II lectin staining in *BmpR1a*^ΔFoxL1+^ mice revealed the scattered expression in goblet cells compared with an orderly staining pattern along the distal crypts in control mice (Figure 4J–L). PNA-lectin staining recognises terminal galactose β1,3 GalNAc residues. Without chemical treatment or enzymatic digestion, PNA-lectin was only detected in the Golgi complex. After desulphation-KOH-sialidase digestion, an equal number of PNA-lectin^+^ goblet cells were found in the control and mutant mice (Figure 4M–O); the peripheral accumulation of lectin in *BmpR1a*^ΔFoxL1+^ mice was similar to that of MUC2.

### 3.5. Impaired BMP Signalling Does Not Affect Telocytes’ Number but Their Localisation toward the crypt axis following Inflammatory Stress

Ultrastructural analysis revealed the classical features of telocytes, i.e., small bodies and long telopodes, in both groups, with *BmpR1a*^ΔFoxL1+^ mice exhibiting more irregularly shaped telopodes than the control (Figure 5A). At higher magnification, TC^FoxL1+^ with loss of BMP-associated signalling were surrounded by densely packed collagen fibrils, suggesting the promotion of secretory activities; their telopodes had more pinched off or shed vesicles than that of the control mice.

We next investigated the TC^FoxL1+^ distribution along the crypts before, during and after an inflammatory flare. Telocytes were defined as Gli1^+^/PDGFRα^+^/CD34^−^ cells. Before treatment, no difference was detected in the TC^FoxL1+^ population, distribution and localisation between both groups (Figure 5B,C). After the acute phase, both control and *BmpR1a*^ΔFoxL1+^ mice presented a decrease in TC^FoxL1+^ population and we found mostly single-labelled (Gli1^+^ or PDGFRα^+^) cells in their stroma (Figure 5B,C). After recovery, both control and *BmpR1a*^ΔFoxL1+^ mice restored their preinjury TC^FoxL1+^ population (Figure 5B,C), but only in control mice did they re-establish their localisation along the colonic epithelial axis. We found that the majority of the TC^FoxL1+^ population was shifted to the upper part of the colonic crypt in *BmpR1a*^ΔFoxL1+^ mice (Figure 5B,D). Following this observation, we investigated the status of myofibroblasts in our model.

Under naive conditions, myofibroblasts were distributed along the crypt vertical axis in both mutant and control mice (Figure 6A,B,G). Upon acute inflammatory stress, myofibroblasts’ presence was exacerbated in both control and *BmpR1a*^ΔFoxL1+^ mice (Figure 6C,D,G). After recovery, we observed an important enduring population of myofibroblasts scattered in the stroma with a strong presence at the bottom of the crypts near the stem cell region in *BmpR1a*^ΔFoxL1+^ mice, whereas the control myofibroblasts population returned to their basal level (Figure 6E–G). Myofibroblasts counts along the crypt vertical axis in the various conditions confirmed a significant enduring myofibroblasts population following recovery in *BmpR1a*^ΔFoxL1+^ mice when compared to controls (Figure 6G).

## 4. Discussion

BMP signalling is involved in the development and homeostasis of the gastrointestinal tract [2,8,10,12]. BMP ligands are widely produced by both the epithelial and mesenchymal compartments [7], and in conjunction with other cascades, BMP maintain the critical balance between cell proliferation and differentiation, thus maintaining gut homeostasis [7,12]. Defective BMP signalling is frequently observed during IBD pathogenesis [2,27]. Recently, we have demonstrated that the targeted disturbance of BMP signalling in mouse TC^FoxL1+^ led to the reprogramming of the stroma, thus initiating neoplasia in the gastric and colonic epithelia [8,10]. However, the precise role of signalling-impaired TC^FoxL1+^ in IBD susceptibility and wound healing remains elusive. Without inflammatory stress, *BmpR1a*^ΔFoxL1+^ mice possessed dysplastic areas with an enlarged mesenchymal compartment and prominent immune cell infiltration [8]. These findings suggest that impaired BMP signalling in TC^FoxL1+^ plays an active role during and after an inflammatory flare; however, they must first be triggered to activate an inflammation cascade. In this study, using different DSS challenges, we revealed a novel role for TC^FoxL1+^ in colon inflammation, resolution and repair. During the acute phase, *BmpR1a*^ΔFoxL1+^ mice demonstrated increased susceptibility to experimental colitis due to disrupted mucus layer integrity and functionality. However, mucosal damage in *BmpR1a*^ΔFoxL1+^ mice resulted in slow mucosal recovery that proved lethal. After recovery, the signalling-impaired TC^FoxL1+^ cells showed abnormal shape and their population were disproportionally found at the top of the crypt with a scarce presence at the bottom near the stem cells and progenitors region (Figure 7). This result indicates that although the number of TC^FoxL1+^ could be important for epithelial recovery, the difference observed regarding the delays in healing in our model does not come from a decrease in TC^FoxL1+^ in the mucosa.

The participation of telocytes in tissue repair has been reported in several organs [28,29] and two primary roles have been postulated: acting as progenitor cells or modulating stem cell activity [30]. Recent studies have shown that mice with PDGFRα^+^ cells lacking R-spondin 3 exhibit increased sensitivity to DSS-mediated inflammation that affects stem cells [11] and that CD34^+^ cells are located in regions of active regeneration, thus influencing the progenitors [4].

In this study, recovery experiments revealed that while TC^FoxL1+^ in control mice resume their natural position along the colonic crypt, myofibroblasts are now more noticeable than TC^FoxL1+^ near the stem cells in *BmpR1a*^ΔFoxL1+^ mice. In recent years, the importance of TC^FoxL1+^ in stem cell niche regulation, via the secretion of WNT [6,11] and BMP factors [7], has been demonstrated. As morphogens, even a small dysregulation in the concentration or gradient of WNT and BMP ligands could affect stem cell survival and cellular fate [7]. Our results suggest that the delocalisation of TC^FoxL1+^ population along the crypt in the mutant following recovery could affect tissue homeostasis, leading to more severe inflammation and delayed healing. Currently, some of the secreted factors of TC^FoxL1+^ have been identified; however, the conducive conditions for TC^FoxL1+^ survival, expansion and homeostasis remain unclear. BMP is fundamental for epithelial cell differentiation [12] and our results suggest that it might not be different for TC^FoxL1+^. In other words, the BMP signalling is most likely required by TC^FoxL1+^ for its optimal functionality in an autocrine/paracrine feedback manner.

Ultrastructural analysis results demonstrated that TC^FoxL1+^ in *BmpR1a*^ΔFoxL1+^ mice were surrounded by a more complex microenvironment than the control mice. Telocytes are involved in mechanical sensing, serve as scaffold platform for stroma elements, and organise the ECM [31]. Thus, their behaviour is altered depending on the nature of the ECM as shown in other tissues [8,31,32,33]. Niculite et al. have demonstrated that in the presence of different enriched matrices, telocytes change their adherence, morphology and telopodes [33]. In addition, extracellular vesicles produced by telocytes have been observed in several tissues [31,32]. Upon the release of these extracellular vesicles and other soluble factors, TC^FoxL1+^ exchange information with the surrounding microenvironment to dictate stem-cell behaviour [6], tissue regeneration [34] and immune cell monitoring [6,35]. We found no significant difference in TC^FoxL1+^ marker expression and localisation along the colon crypts under naive conditions; however, upon acute inflammatory stress, the TC^FoxL1+^ population decreased, whereas that of myofibroblasts increased in both control and mutant mice. This reduction in TC^FoxL1+^ has been previously described in ulcerative colitis [36] and has been hypothesised to facilitate uncontrolled myofibroblast proliferation, thus increasing ECM protein deposition. Excessive ECM production contributes to the disruption of tissue homeostasis and the development of fibrosis in inflammatory diseases [36]. The increase in myofibroblasts seen in both groups during the acute phase was maintained only in the stroma of *BmpR1a*^ΔFoxL1+^ mice following recovery, suggesting that apoptosis or trans-differentiation from myofibroblast to fibroblast, normally expected during restitution, does not occur [37]. Hence, this indicates that TC^FoxL1+^ with impaired BMP signalling influence the homeostasis of the surrounding stromal cells, possibly through secretion of soluble factors or extracellular vesicles [6,34,35]. This paracrine signalling with the surrounding stroma cells will be further developed in upcoming studies.

The dominant phenotype of *BmpR1a*^ΔFoxL1+^ mice was the presence of an irregular colonic mucus, more specifically in the barrier layer. Upon an inflammatory insult, bacteria infiltrated the epithelial barrier of mutant mice as shown in the MUC2-deficient mouse model [38]. Recovery experiments demonstrated that in *BmpR1a*^ΔFoxL1+^ mice, there was delayed colonic mucosa healing, along with a decreased number of goblet cells, denuded epithelial regions and structural problems in the mucus layers, long after the control group had recovered from the insult. We observed no difference in the expression of genes related to cell-fate decisions in both the secretory and absorbent lineages between control and mutant mice, indicating that TC^FoxL1+^ with impaired BMP signalling do not influence progenitor cells that directly affect cell determination. However, reduction in the maturation gene *Klf4* in *BmpR1a*^ΔFoxL1+^ mice suggested a possible defect in goblet cell maturation and functionality following loss of BMP signalling in TC^FoxL1+^.

We focused on the analysis of goblet cell functionality, mucus synthesis and production in our mouse model. Our ultrastructural analysis results provide new insights on the vesicular composition of goblet cells and the expression of apical glycocalyx. *BmpR1a*^ΔFoxL1+^ mice exhibited alterations in the epithelial surface glycocalyx and goblet cell vesicles without heterogeneity. Furthermore, *Muc2* and *Muc4* mRNA expression significantly increased, suggesting a shift in mucin ratios, which could support the deregulation observed in vesicle diversity in mutant mice. In addition, *BmpR1a*^ΔFoxL1+^ mice had a significant increase in the expression of genes such as *Arg2* and *Fut2*, affecting biosynthesis, final structure and functionality of colonic mucins [14,39]. Anterior Gradient 2 (AGR2) has been associated with MUC2 biosynthesis, particularly in its folding, trafficking and assembly [39]. Meanwhile, fucosyltransferase 2 (FUT2) is involved in mucin maturation. Finally, the abnormal pattern of the MUC2 protein within the goblet cell vesicles in *BmpR1a*^ΔFoxL1+^ mice suggests an aberrant glycosylation pattern [40]. Mucins have post-translational modifications that affect their functionality [41]. Altogether, these findings led us to analyse components related to mucus structure, such as glycosylation patterns, that could yield a low mucus quality and support the increased susceptibility to experimental colitis observed in *BmpR1a*^ΔFoxL1+^ mice.

*O*-glycosylation of mucins is initiated by the addition of GalNAc to the hydroxyl groups of serine or threonine to form the Tn antigen [42]. Further steps lead to different core structures containing galactose and terminal residues, such as fucose or sialic acid [43,44]. These terminal residues are known to participate in gut microbiota homeostasis [45,46]. Here, in *BmpR1a*^ΔFoxL1+^ mice, UEA-1 and SNA lectin had reduced fucose and α2,6-linked sialic acid residues. Similar to previous studies [21,47], we only detected galactose residues after desulphation and enzymatic digestion, which can be attributed to further glycosylation or sulphation of the T antigen [21,47]. These results suggest that *BmpR1a*^ΔFoxL1+^ mice exhibit a defective glycosylation pathway, leading to the production of immature mucin. Hence, the reduction in these residues could not only impair mucus quality and promote bacterial accessibility to the epithelium [48], but also shape the gut microbial community and its relationship with the host [49]. Indeed, because glycans serve as carbon energy sources [46] and attachment sites for the resident microbiota, future studies will be required to analyse the microbiota composition in both groups.

In summary, we have clearly demonstrated the involvement of TC^FoxL1+^ in the regulation of the healing functions of the colonic mucosa during and after inflammation. *BmpR1a*^ΔFoxL1+^ mice showed abnormal mucus quality, and following inflammation, they had increased susceptibility to experimental colitis and delayed healing. Our results suggest that cell–cell paracrine communication or direct interactions between goblet cells and TC^FoxL1+^ are essential for maintaining the optimal functionality of the secretory cells. In addition, these results revealed that BMP signalling in TC^FoxL1+^ is key for the regulation of its own homeostasis and communication functionality. Our findings provide new insights into the roles of TC^FoxL1+^ in the regulation of epithelial homeostasis beyond the stem cell niche.

## Figures and Tables

**Figure 1 cells-10-02954-f001:**
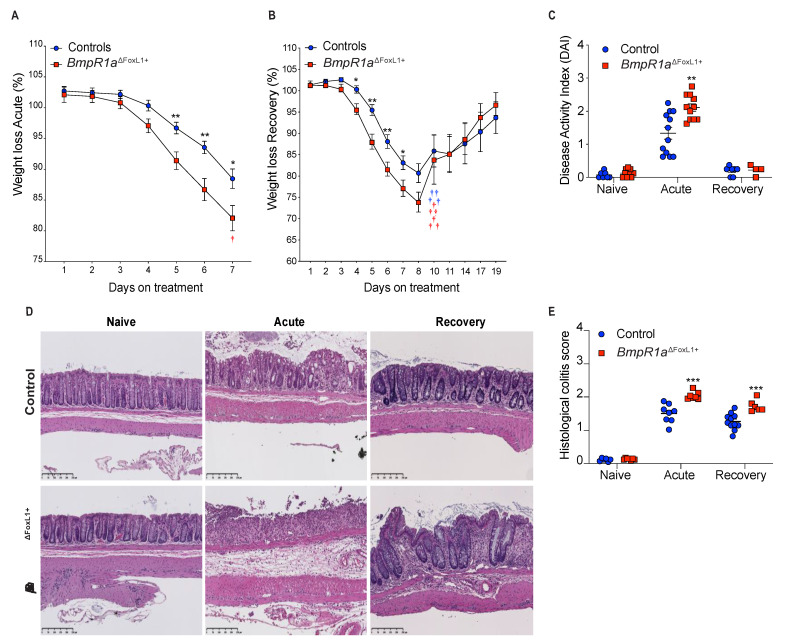
Loss of BMP signalling in telocytes predisposes mice to severe experimental colitis. (**A**) *BmpR1a*^ΔFoxL1+^ mice (red squares) had greater weight loss than control mice (blue circles) following the induction of acute experimental colitis and (**B**) the subsequent recovery phase (*N* = 10). (**C**) DAI analysis revealed a significant DAI increase in mutant mice only during the acute phase. (**D**) H&E staining shows normal mucosa architecture under naive condition in all groups. During the acute phase, control mice presented crypt erosion but the epithelial layer was preserved while *BmpR1a*^ΔFoxL1+^ mice presented a complete loss of their epithelial sheet. After recovery, control mice showed a restored mucosa, whereas *BmpR1a*^ΔFoxL1+^ mice still presented immune infiltration and regions with crypt distortion. (**E**) Histological scores reveal a significant increase in mutant mice in both treatments. Data are expressed as the mean ± SEM (*N* = 19–20). * *p* < 0.05; ** *p* < 0.01; *** *p* < 0.001 analysed by Mann–Whitney test. Scale bar: 250 μm. † Animals that died during the experiment.

**Figure 2 cells-10-02954-f002:**
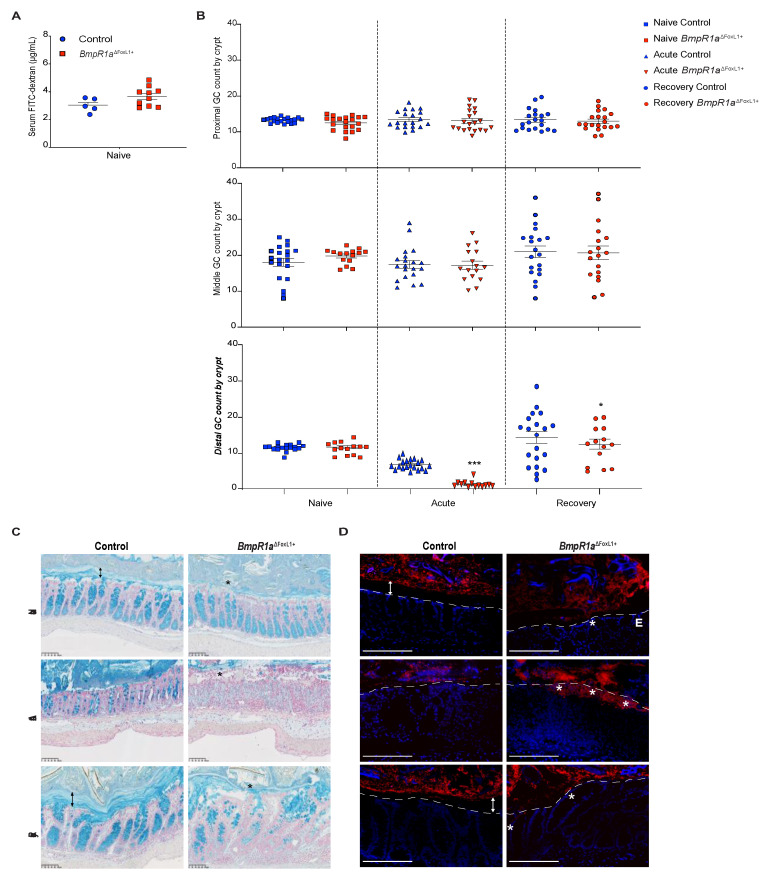
Impaired barrier layer in *BmpR1a*^ΔFoxL1+^ mice resulted in susceptibility to bacterial mucosal invasion. (**A**) No differences in concentration of FITC-dextran were observed in serum of controls and *BmpR1a*^ΔFoxL1+^ mice (*N* = 5–10). (**B**) Statistical analysis showed a significant decrease in the number of acidic mucin-positive cells in mutant mice compared with the control, only in the distal colon after both (acute and recovery) treatments; no significant modulation was observed in other colonic sections (*N* = 4; *n* = 20). After the acute phase, there was a 0.2-fold decrease in goblet cells, and 0.8-fold decrease following recovery. (**C**) Under naive conditions, a well-preserved barrier layer (double black arrow) was observed in control mice, whereas in *BmpR1a*^ΔFoxL1+^ mice a discontinuous (black asterisk) and thinner barrier layer was observed (*N* = 8). During the acute phase, all mice showed partial erosion of the barrier layer with *BmpR1a*^ΔFoxL1+^ mice presenting more advanced erosion. After recovery, *BmpR1a*^ΔFoxL1+^ mice still presented a thinner barrier layer while control mice improved. (**D**) Before treatment, bacteria (red) were localised only in the outer mucus layer and not found in the inner layer (double white arrows) in control mice; during the acute phase, bacteria were in contact with the epithelium but not detected in the inner mucosa. In *BmpR1a*^ΔFoxL1+^ mice, during the acute phase, large amounts of bacteria (white asterisks) invaded the colonic mucosa. Following recovery, the control mice restored the sterility of the barrier layer (double white arrow), whereas mutant mice still presented bacteria in contact with the epithelium. Nuclei were counterstained with DAPI (blue). Epithelium is delimited by the white discontinuous line. E: Epithelium. Scale bars: 250 μm (**C**); 105 μm (**D**). * *p* < 0.05; *** *p* < 0.001 analysed by Mann–Whitney test.

**Figure 3 cells-10-02954-f003:**
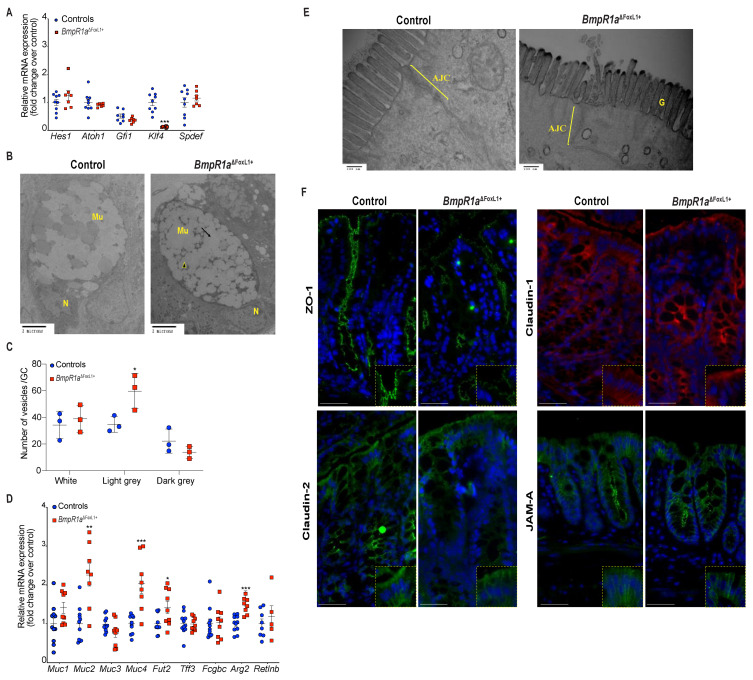
*BmpR1a*^ΔFoxL1+^ mice displayed abnormalities in goblet cell vesicles and presented dysregulated mucus maturation and structural gene expression. (**A**) Relative mRNA levels of the colonocyte/secretory lineage specification *Hes1* and *Atoh1*, along with committed-secretory cell toward the goblet fate or maturation *Gfi1*, *Spdef and Klf4*. A significant decrease was observed for *Klf4* mRNA levels in mutant mice while other mRNAs were not modulated between mutant and control mice (*N* = 7–9). (**B**) Ultrastructural analysis of the goblet cells revealed vesicles without mucin diversity, disturbed morphology in *BmpR1a*^ΔFoxL1+^ mice (*N* = 4). (**C**) Vesicles pattern quantification revealed a significant increase in light grey vesicles in mutant goblet cells when compared to controls (*N* = 3). (**D**) *BmpR1a*^ΔFoxL1+^ mice showed a significant increase in *Fut2*, *Muc2*, *Muc4* and *Agr2* mRNA levels in mutant mice. (**E**) Ultrastructural analysis of the glycocalyx and the apical junctional complex (yellow bracket) revealed that the latter was not modified between both groups and showed loss of glycocalyx in *BmpR1a*^ΔFoxL1+^ mice (*N* = 4). (**F**) Immunofluorescence against ZO-1, claudin-1 and 2 as well as JAM-A revealed no modulation in the tight junction complex proteins between both groups (*N* = 3). (**C**) 2-way ANOVA test. Data are presented as the mean ± SD. (**D**) Mann–Whitney test. N: Nucleus; Mu: Mucin vesicles; G: Glycocalyx; AJC: Apical junctional complex. Fold-change was normalised to that of TATA box protein (*TBP*) used as housekeeping gene (*N* = 10). * *p* < 0.05; ** *p* < 0.01; *** *p* < 0.001 analysed by Mann–Whitney test.

**Figure 4 cells-10-02954-f004:**
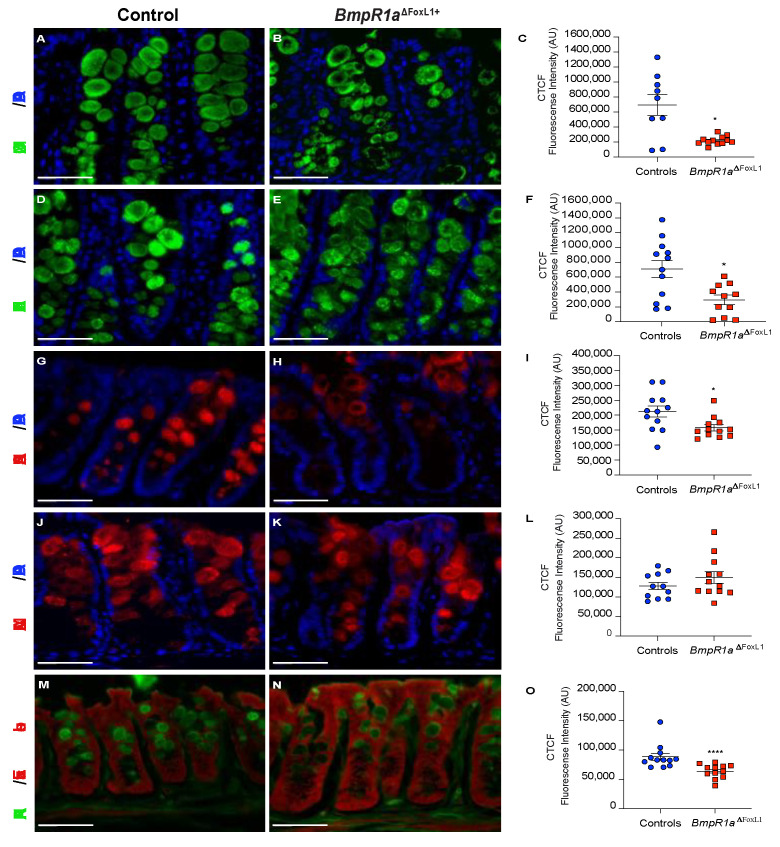
Loss of BMP signalling in telocytes promotes abnormal MUC2 localisation within secretory vesicles and reduces mucin maturation. (**A**) Control mice presented well-filled mucin vesicles (**B**) whereas *BmpR1a*^ΔFoxL1+^ mice showed abnormal mucin (green) deposition at the periphery of the vesicles. (**C**) Intensity measurements across goblet cells (middle vs. periphery) validated the peripheral deposition of MUC2 in the mutant mice. (**D**) Control mice presented normal levels of mucin fucosylation (green) while (**E**) *BmpR1a*^ΔFoxL1+^ mice revealed a decrease in mucin fucosylated residues. (**F**) Intensity measurements across goblet cells validated decreased mucin fucosylated residues in mutant mice. (**G**) Control mice presented homogenous expression pattern for sialic acid modifications (red) in mucins whereas (**H**) *BmpR1a*^ΔFoxL1+^ mice presented a peripherical deposition in the vesicles. (**I**) Intensity measurements across goblet cells validated reduction in patterns of sialic acid modifications in the mutant mice. (**J**) Control mice revealed an orderly staining pattern for sialic acid residues (red) in vesicles while (**K**) *BmpR1a*^ΔFoxL1+^ mice showed scattered distribution and reduced expression. (**L**) Intensity measurements across goblet cells confirmed no modulation in patterns of sialic acid residues between both groups. (**M**) Control mice presented normal vesicles pattern for terminal galactose β1,3 GalNAc residues (green) whereas (**N**) *BmpR1a*^ΔFoxL1+^ mice showed accumulation of the residues at the periphery of vesicles. (**O**) Intensity measurements across goblet cells confirmed a reduction in terminal galactose β1,3 GalNAc residues in mutant mice. Nuclei were counterstained with DAPI (blue); (**M**,**N**) were counterstained with Evan’s blue (red) (*N* = 4). Scale bar: 450 μm. * *p* < 0.05; **** *p* < 0.0001 analysed by Mann–Whitney test.

**Figure 5 cells-10-02954-f005:**
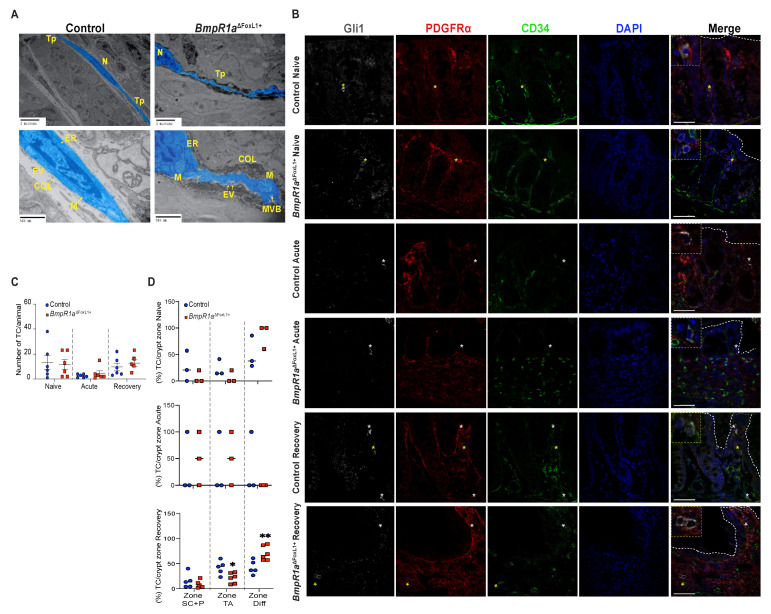
Loss of BMP signalling leads to irregularly shaped telocytes under naive conditions, whereas inflammatory challenge and recovery do not affect its number among groups. (**A**) Ultrastructural analysis of telocytes (blue) present classical features for these cells with small bodies and long telopodes in control and *BmpR1a*^ΔFoxL1+^ mice. Presence of irregularly shaped telopodes were observed in *BmpR1a*^ΔFoxL1+^ mice. Higher magnification demonstrated increased secretory activities from TC^FoxL1+^ in *BmpR1a*^ΔFoxL1+^ mice as shown by presence of increased collagen fibrils deposition and extracellular vesicles compared with the control. (**B**) Under naive condition, no difference was observed in TC^FoxL1+^ population numbers (Gli1^+^/grey, PDGFRα^+^/red, CD34^−^/green) (yellow asterisks) nor in localisation in regard to the epithelium (white discontinuous line) in both groups. Upon acute phase, TC^FoxL1+^ population was reduced in both control and *BmpR1a*^ΔFoxL1+^ mice and we detected mostly CD34^+^ or PDGFRα^+^ cells (white asterisks). Following recovery, TC^FoxL1+^ population recovered in both groups. *BmpR1a*^ΔFoxL1+^ mice showed a change of its TC^FoxL1+^ population in regard to their expected positions along the colonic crypt vertical axis. Higher magnification shows TC^FoxL1+^ (yellow square), CD34^+^ or PDGFRα^+^ cells (white square). (**C**) Total number of TC^FoxL1+^ in colon before, during and after inflammatory stress were counted and no significant difference was observed between groups (*N* = 6). (**D**) TC^FoxL1+^ frequency along the colonic vertical axis separated in three cell zones revealed a clear change in TC^FoxL1+^ population along the axis during the recovery phase in *BmpR1a*^ΔFoxL1+^ mice compared to controls. No modulation in TC^FoxL1+^ population along the colonic epithelial axis was found under naive or acute condition between both groups. (**C**) Mann–Whitney test. (**D**) 2-way ANOVA test. Data are presented as the mean ± SD (*N* = 3). Scale bar: 50 μm. N: Nucleus; Tp: Telopodes; EV: Extracellular vesicles; ER: endoplasmic reticulum; COL: Collagen fibrils; M: Mitochondria; MVB: Multivesicular bodies; SC: Stem Cells; P: Progenitor Cells; TA: Transit Amplifying; Diff.: Differentiated cells.

**Figure 6 cells-10-02954-f006:**
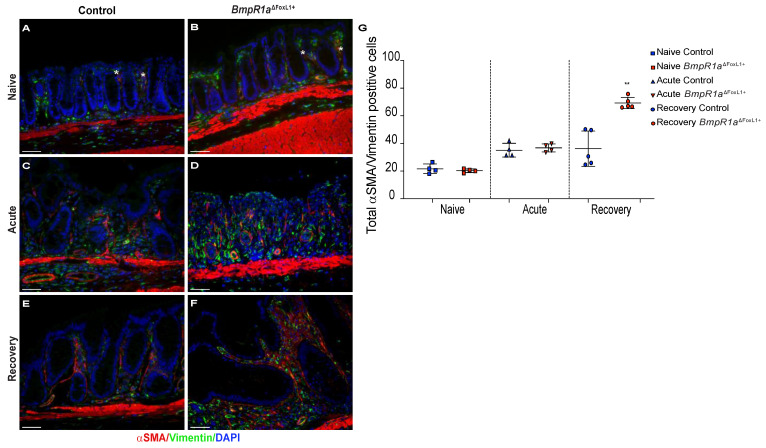
Myofibroblasts are localised near stem cells and along the epithelial sheet upon recovery in *BmpR1a*^ΔFoxL1+^ mice. (**A**) Under naive condition, control mice demonstrated the basal myofibroblast (white asterisks) population (**B**) which was comparable in *BmpR1a*^ΔFoxL1+^ mice. (**C**) During the acute phase, control mice exhibited an increase in myofibroblasts (white asterisks) in regard to the naive condition with (**D**) with a similar increase in *BmpR1a*^ΔFoxL1+^ mice. (**E**) After recovery, control mice showed a return to the basal level in myofibroblast population seen under naive condition (**F**) while *BmpR1a*^ΔFoxL1+^ mice still presented a strong presence of myofibroblasts scattered in their mucosa with these cells also found in the region associated with the stem cell niche and progenitor cells. (**G**) Myofibroblasts count along the crypt axis revealed a 1.9-fold increase in myofibroblasts in mutant mice compared to controls after the recovery phase. The myofibroblast population was found to be similar in both groups in the naive condition and during the acute phase. Scale bar: 50 μm. Mann–Whitney test (*N* = 4–6) ** *p* < 0.01.

**Figure 7 cells-10-02954-f007:**
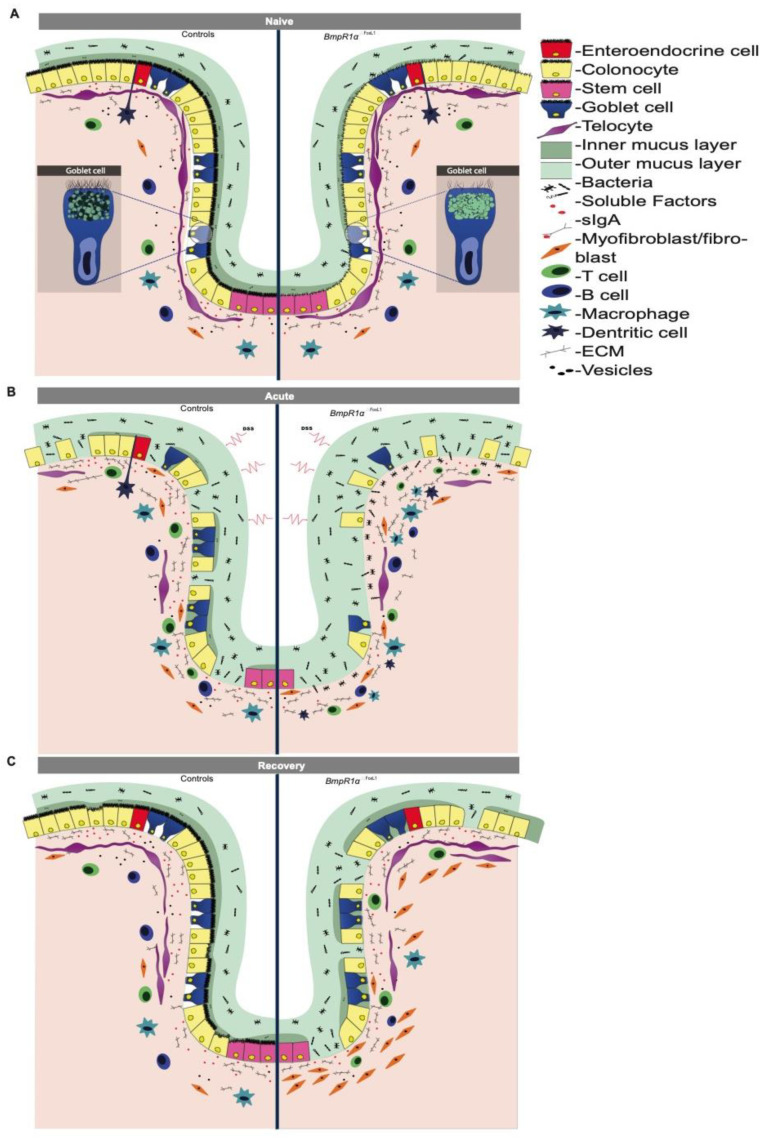
Schematic representation of IBD pathogenesis events in colonic mucosa with impaired BMP-signalling telocytes. (**A**) Under naive condition *BmpR1a*^ΔFoxL1+^ mice showed reduced diversity in goblet cell vesicles, low glycocalyx coating and a thinner and discontinuous barrier layer with bacteria in close contact with the non-inflamed epithelium. *BmpR1a*^ΔFoxL1+^ mice exhibited TC^FoxL1+^ population with an irregular shape and surrounded by collagen fibers. (**B**) During inflammation, a complete depletion of goblet cells was observed in *BmpR1a*^ΔFoxL1+^ mice and a significant number of bacteria invaded the inflamed colonic mucosa. TC^FoxL1+^ population was decreased in both groups. A substantial presence of myofibroblasts were found in *BmpR1a*^ΔFoxL1+^ mice compared to controls. (**C**) After the recovery, mucosa in control mice recovered from the insult while in *BmpR1a*^ΔFoxL1+^ mice it presented delayed wound healing. In *BmpR1a*^ΔFoxL1+^ mice, TC^FoxL1+^ population was found to be displaced along the colonic crypt with more TC^FoxL1+^ found at the top and less at the bottom. Myofibroblast-like cells (vimentin^+^; αSMA^+^) were still strongly present in the stroma. Myofibroblast-like cells were found to be scattered in the stroma with a robust presence at the base of the crypt.

## Data Availability

Data is contained within the article.

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
