# Peer review of "Altered Mucus Barrier Integrity and Increased Susceptibility to Colitis in Mice upon Loss of Telocyte Bone Morphogenetic Protein Signalling"

_cells, 2021, doi:10.3390/cells10112954_

Round 1

Reviewer 1 Report

Please see attached Word document.

Reviewer 2 Report

In this study, the authors investigated the role of BMP signaling in FoxL1+-Telocytes (TCFoxL1+) in susceptibility to Dextran Sulfate Sodium (DSS)- induced acute intestinal damage and mucosal wound healing during recovery phase. By using mice with loss of BMP signaling pathway in TCFoxL1+ cells, authors provide evidence that loss of BMP signaling in TCFoxL1+ cells resulted in alteration of mucin production by goblet cells which was associated with defective mucus layer formation and increased bacterial invasion as compared with control mice. Defective BMP signaling in TCFoxL1+ cells worsened mucosal damage after DSS treatment and delayed mucosal healing. Interestingly, BMP signaling in TCFoxL1+ is shown to play a key role in maintaining mesenchymal cells in close contact with the epithelium and controlling the number of myofibroblasts in the tissue. Overall, the manuscript is well presented, and this study provides new insights into mesenchymal cell function in the pathogenesis of IBD

While most of the data supports the conclusions, the study would be strengthened with few additions.

  • It would be informative to know whether the intestinal permeability is increased in deficient mice after acute DSS-induced colitis and recovery phase. Please provide the data of in vivo permeability for all the conditions.
  • In figure 3B, the AJC between neighboring cells is not shown for the control. Please replace the representative EM image for the control. Also, immunofluorescence images showing the localization of well-accepted AJC proteins such as ZO-1, claudins or occludin are needed to validate that AJC composition is not altered between groups.
  • In figure 5B, please improve the signal/quality of the immunofluorescence images.

Minor Concerns:

- Remove duplicate words “of goblet cells” in line 182.

- Figure legends are unreadable in figures 1D, 2C and 4 and 6.

Author Response

We thank Reviewer 2 for the provided comments. All specific comments to the authors

were taken into consideration.

1. It would be informative to know whether the intestinal permeability is increased in deficient mice after acute DSS-induced colitis and recovery phase. Please provide the data of in vivo permeability for all the conditions.

We agree that demonstration of in vivo permeability for all the conditions could provide some additional information. To do so, we would need to generate additional mouse colonies, wait until they reach 90 days-old and then perform the recovery experiments for 19 days (5 days for the acute phase and 14 days of recovery). Unfortunately, we are unable to perform such an exhaustive experiment within the 10 days limit for this review process.

2. In figure 3B, the AJC between neighboring cells is not shown for the control. Please replace the representative EM image for the control. Also, immunofluorescence images showing the localization of well-accepted AJC proteins such as ZO-1, claudins or occludin are needed to validate that AJC composition is not altered between groups.

TJ proteins were analyzed by immunofluorescence between groups and are now shown in figure 3F.

3. In figure 5B, please improve the signal/quality of the immunofluorescence images. 

These images are the best we can provided at this time. They were acquired on a biphoton microscope. However, size limitation when we uploaded the manuscript limit us in the best image quality we can provide to the reviewers.

Minor Concerns:

The duplicate “of goblet cells” in line 182 has been removed.

Figure legends are unreadable in figures 1D, 2C and 4 and 6

 These figure legends have been modified.

Best regards,

Nathalie Perreault PhD

Reviewer 3 Report

In the present basic science article Reyes Nicolas et al investigated the effect of BMP loss in subepithelial telocytes on chemically induced colitis in mice. They found that such loss alters mucus composition, which becomes less effective in the protection against colitis. Additionally, BMP loss led to alteration of telocyte shape and distribution as well as in stem cell niche function.

This is an excellent paper and I have no criticism to raise.

Author Response

We thank Reviewer 3 for the time provided in reading our manuscript and his enthusiasm in recommending acceptance.

Best regards,

Nathalie Perreault, PhD

Round 2

Reviewer 1 Report

The authors have adequately addressed my concerns; this manuscript is significantly improved over the original.